# The β Isoform of Human ATP-Binding Cassette B5 Transporter, ABCB5β, Localizes to the Endoplasmic Reticulum

**DOI:** 10.3390/ijms242115847

**Published:** 2023-10-31

**Authors:** Adriana María Díaz-Anaya, Louise Gerard, Martine Albert, Jean-François Gaussin, Marielle Boonen, Jean-Pierre Gillet

**Affiliations:** 1Laboratory of Molecular Cancer Biology, URPhyM, NARILIS, University of Namur, 5000 Namur, Belgium; adriana.diazanaya@unamur.be (A.M.D.-A.); louise.gerard@unamur.be (L.G.); 2Laboratory of Intracellular Trafficking Biology, URPhyM, NARILIS, University of Namur, 5000 Namur, Belgiumjean-francois.gaussin@unamur.be (J.-F.G.)

**Keywords:** ABCB5β, ABC transporters, melanoma, subcellular localization

## Abstract

ABCB5β is a member of the ABC transporter superfamily cloned from melanocytes. It has been reported as a marker of skin progenitor cells and melanoma stem cells. ABCB5β has also been shown to exert an oncogenic activity and promote cancer metastasis. However, this protein remains poorly characterized. To elucidate its subcellular localization, we tested several anti-ABCB5 antibodies and prepared several tagged *ABCB5β* cDNA constructs. We then used a combination of immunofluorescence and biochemical analyses to investigate the presence of ABCB5β in different subcellular compartments of HeLa and MelJuSo cell lines. Treatment of the cells with the proteasome inhibitor MG132 showed that part of the population of newly synthesized ABCB5β is degraded by the proteasome system. Interestingly, treatment with SAHA, a molecule that promotes chaperone-assisted folding, largely increased the expression of ABCB5β. Nevertheless, the overall protein distribution in the cells remained similar to that of control conditions; the protein extensively colocalized with the endoplasmic reticulum marker calnexin. Taken together with cell surface biotinylation studies demonstrating that the protein does not reach the plasma membrane (even after SAHA treatment), the data indicate that ABCB5β is a microsomal protein predominantly localized to the ER.

## 1. Introduction

ABCB5 is a member of the ATP-binding cassette (ABC) transporter superfamily [1]. Humans have 48 ABC genes encoding 44 membrane transporters classified into five families (A, B, C, D, and G). These transporters are composed of two symmetric halves, each including a transmembrane domain (TMD) that facilitates substrate export (for most of them) or import (for ABCA4 and ABCD4) and a nucleotide-binding domain (NBD) responsible for ATP binding and hydrolysis. They are expressed either as full transporters containing two nonidentical halves within a single polypeptide or as half transporters that have to homo- or heterodimerize to be functional. There are also four non-transporter ABC proteins that belong to families E and F. They exist as twin NBDs without any TMDs [2].

The ABC transporters translocate a wide variety of endogenous and xenobiotic substrates across membranes, including peptides, polysaccharides, glutathione conjugates, antibiotics, and anticancer drugs, among many others [3]. They have been extensively studied for their roles in cancer multidrug resistance and, more recently, in tumorigenesis [4,5]. Moreover, mutated ABC genes cause several monogenic diseases (including cystic fibrosis) and may increase susceptibility to complex diseases such as coronary artery disease and Alzheimer’s disease [6,7]. 

*ABCB5* is one of the most frequently mutated genes in melanoma [8,9]. This is the deadliest type of skin cancer, which represents a significant challenge in clinical oncology due to intrinsic and acquired mechanisms of resistance to treatments [10,11]. ABCB5 was also reported to be a marker of skin progenitor cells [12] and melanoma stem cells [13] and to mediate anthracycline resistance [14,15]. 

*ABCB5* encodes a full transporter (ABCB5FL) and a half transporter (ABCB5β). ABCB5FL was cloned from testis [15] and ABCB5β was cloned from melanocytes [12]. There are also many additional transcript variants, including ABCB5α, but they are too short to form functional transporters [16]. ABCB5FL was shown to confer low-level multidrug resistance. The drugs transported by this full-length transporter include anthracyclines, taxanes, vinca-alkaloids, and epipodophyllotoxin [15,17]. Drugs transported by ABCB5β have not been identified thus far, but only a limited number of molecules have been tested. ABCB5β might transport some other drugs and/or act as a drug-efflux transporter as a heterodimer.

It has been shown that several mutations in *ABCB5* promote the proliferation and invasive capacities of melanoma cells [9]. It has also been reported that melanoma cells with a high *ABCB5* expression exhibit an elevated metastatic potential both in vitro and in vivo [18]. However, the underlying mechanisms and to what extent both ABCB5 isoforms interplay have yet to be unraveled. Intriguingly, Chen et al. reported that *ABCB5FL* was undetectable in a melanotic cDNA library, in contrast to α and β isoforms. These were also found enriched in melanoma cells compared to several normal tissues [16]. Hence, we decided to focus on the ABCB5β isoform that, in contrast to the α isoform, has a predicted transport function. Our aim here was to answer a question that remains unresolved, and that is an important requirement in the search for the molecular role of this protein, i.e., its subcellular localization. To address this, we tested anti-ABCB5 antibodies and prepared several tagged *ABCB5β* cDNA constructs. We then used a combination of immunofluorescence analyses to investigate the presence of ABCB5β in different cell compartments of HeLa (cervical cancer) and MelJuSo (melanoma) cell lines.

## 2. Results

### 2.1. Western Blotting Signals Detected with Three Different Commercial Anti-ABCB5 Antibodies Do Not Decrease after Knock-Down of ABCB5 in MelJuSo Cells

First, we tested whether we could detect the endogenous expression of ABCB5β at the protein level in MelJuSo cells. We used three different polyclonal anti-ABCB5 antibodies commercially available. The Abcam antibody was raised against the first 99 amino acids of ABCB5β, whereas the Rockland and Atlas antibodies recognize overlapping epitopes located between amino acids 192–208 and 145–234, respectively (Appendix A). All of them detected human ABCB5β by Western blotting when overexpressed in membrane vesicles of high-five insect cells (Figure 1A). To assess their specificity for the detection of endogenously expressed ABCB5, we silenced *ABCB5* expression at the mRNA level in MelJuSo cells using a pool of four siRNAs and collected the samples up to 96 h post-transfection (Figure 1B). The four siRNAs recognize sequences that are found in both *ABCB5FL* and *ABCB5β*. However, although RT-qPCR revealed a ≥ 90% decrease in *ABCB5* expression up to 72 h post-transfection (most likely of the β isoform since FL is barely expressed, if at all, at the basal level, according to Chen et al. [16]), none of the signals detected by Western blotting close to the expected molecular weight for ABCB5β (90 kDa) decreased after siRNA transfection, nor any other bands for that matter (Figure 1C).

We also tested the three commercial antibodies in an immunofluorescence application (i.e., a method that detects proteins in native or near-native conformation, in contrast to Western blotting using SDS-PAGE). Only the Rockland anti-ABCB5 antibody gave rise to a detectable fluorescent signal, which did not decrease in MelJuSo cells transfected with siRNAs targeting *ABCB5* (Appendix A).

Taken together, these results highlight that several commercial antibodies can detect large amounts of ABCB5β, e.g., after overexpression in high-five insect cells, but fail to detect the protein expressed endogenously in melanoma cells. Of note, similar results were reported by Louphrasitthiphol et al., who tested three additional commercial anti-ABCB5 antibodies to analyze ABCB5 expression in the 501 MEL cell line [19]. These data motivated the use of tagged-ABCB5 constructs to address the localization of ABCB5β proteins in the cell.

### 2.2. GFP-ABCB5β Localizes to the Endoplasmic Reticulum in HeLa and MelJuSo Cells

As we cannot study the subcellular localization of the endogenous ABCB5β with commercial antibodies, we decided to engineer GFP-tagged *ABCB5β* constructs, with the GFP either fused at the N-terminus or the C-terminus of the protein (GFP-ABCB5β or ABCB5β-GFP, respectively). The chimeric proteins were then expressed in HeLa or MelJuSo cells using a pcDNA3.1(+) plasmid with a conventional CMV promoter. As shown in Figure 2A, a protein band with a molecular mass of approx. 120 kDa was detected by Western blotting 48 h post-transfection of the GFP-ABCB5β construct in HeLa and MelJuSo cells, using an anti-GFP antibody. This band corresponds to the molecular weight of ABCB5β (90 kDa) combined with the eGFP tag (~30 kDa). ABCB5β-GFP (C-ter) was also detected but at a very low level.

Treatment of the cells with the proteasome inhibitor MG132 increased ABCB5β-GFP expression, though it remained quite low in MelJuSo cells (Figure 2B). Of note, this treatment also increased the amount of N-terminally tagged GFP-ABCB5 in both cell types, suggesting partial degradation of the newly synthesized ABCB5β protein population.

Next, we examined the subcellular localization of GFP-tagged ABCB5β in HeLa and MelJuSo cells using confocal microscopy. Although transfection efficiency was quite low, the N-terminally tagged protein could be detected, by contrast to the C-terminally tagged protein. Since ABCB5 is highly homologous with ABCB1, which is primarily located at the plasma membrane of cells but is also found in endosomes, lysosomes, endoplasmic reticulum, and the Golgi apparatus [20,21], we analyzed putative co-localization between GFP-ABCB5β and markers for these organelles.

In the HeLa cell line, the majority of ABCB5β exhibited cytoplasmic localization within an extensive tubular network, which is a characteristic feature of the endoplasmic reticulum (ER) (Figure 3). The protein exhibited a similar distribution in MelJuSo cells.

Interestingly, we detected co-localization in both cell types with the endoplasmic reticulum marker calnexin (Figure 3A,E for higher resolution micrographs obtained with a Zeiss LSM900 with Airyscan 2 confocal microscope), but with none of the other markers analyzed: GM130 for cis-Golgi, TGN46 for Trans-Golgi, and LAMP1 for late endosomes/lysosomes (Figure 3B–D). We also analyzed a marker of melanosomes (lysosome-like organelles only found in melanoma cells) in MelJuSo cells but did not find any co-localization (Appendix A).

Keeping in mind that a large GFP tag may sometimes result in protein mislocalization, we conducted a series of controls and additional experiments. We notably studied the localization of ABCB9, another half transporter from the same subfamily, by using a GFP-tagged ABCB9 protein. As a half-transporter, ABCB9 and ABCB5β have similar molecular weights (84 kDa and 90 kDa, respectively). Importantly, adding a GFP tag at the N-ter position of ABCB9 did not prevent the transporter from reaching lysosomes, i.e., its reported residence site in the cells [22]. Indeed, GFP-ABCB9 was found to colocalize with LAMP1 (Figure 3F).

### 2.3. HA-ABCB5β Localizes to the Endoplasmic Reticulum in HeLa and MelJuSo Cells

We engineered an additional construct in which ABCB5β was fused to a smaller tag in the N-ter position. Hemagglutinin (HA) is only composed of nine amino acids (YPYDVPDYA). No signal overlap was detected between this tagged protein and other organelle markers, including GM130 for cis-Golgi, TGN46 for trans-Golgi, LAMP1 for lysosomes (Figure 4A–C), and anti-melanoma antibody for melanosomes in MelJuSo (Appendix A). GFP-ABCB5β and HA-ABCB5β co-localized when co-transfected, suggesting that these chimeric proteins exhibit the same localization (Figure 4D). Indeed, when we conducted colocalization analyses as described above, we only observed colocalization between the HA-ABCB5β and the ER marker calnexin (Figure 4E for higher resolution).

As an additional control, we transferred the HA-ABCB5β construct from the pcDNA3.1 plasmid used in previous experiments, i.e., a plasmid that contained a high expression CMV promoter, to a plasmid containing a lower PGK expression promoter. We validated the decreased expression with the PGK promoter by Western blotting (Appendix A). GFP-ABCB5β and pPGK-HA-ABCB5β colocalized when co-transfected, suggesting that these chimeric proteins exhibit the same localization as well (Figure 5A). Again, we only found colocalization between the HA-ABCB5β protein (expressed from the low-expression plasmid) and the endoplasmic reticulum marker calnexin (Figure 5B). No signal overlap was detected between this tagged protein and the other organelle markers (Figure 5C–E). Lastly, we conducted a colocalization analysis with another ER marker, the tail-anchored TA-GFP, which is an ER membrane protein. We also found an important co-distribution with pPGK-HA-ABCB5β (Appendix A).

Taken together, these results support that ABCB5 resides, to a large extent, in the ER under basal conditions.

### 2.4. ABCB5β Remains Localized in the ER after Treatment of the Cells with SAHA

Since part of the population of newly synthesized ABCB5β is degraded by the proteasome system (Figure 2), we also investigated whether the treatment of transfected cells with a small molecule that promotes chaperone-assisted folding, i.e., SAHA (suberoylanilide hydroxamic acid), would modify ABCB5β expression and/or localization in the cells. It has notably been reported that SAHA increases the presence of the ΔF508 mutant of CFTR (ABCC7), which is largely misfolded under basal conditions at the plasma membrane (i.e., the residence site of wild-type CFTR) [23].

Interestingly, the incubation of HeLa cells with 2.5 or 5 µM SAHA largely increased the expression of HA-ABCB5β (by 13- and 25-fold, respectively), consistent with an enhancement of its folding (Figure 6A). Nevertheless, the overall protein distribution in the cells remained similar to that of control conditions; the protein extensively colocalized with the endoplasmic reticulum marker calnexin (Figure 6B).

A few cells exhibited ABCB5β labeling at the cell periphery (see arrows in Figure 6B,C; N.B. cell limits were highlighted with an actin staining in panel C). Considering that ABCB1, a protein with 70% sequence similarity with ABCB5, localizes at the plasma membrane, we biotinylated all cell surface proteins and assessed the presence of HA-ABCB5β among them by Western blotting (Figure 6D). However, the fraction of total HA-ABCB5β proteins recovered in the biotinylated fraction was almost null (0.3 ± 0.6%) and did not increase after treatment with SAHA (marginal fractions of 0.4 ± 0.5% and 0.3 ± 0.5% of total HA-ABCB5β proteins were biotinylated after treatment with 2.5 or 5 µM of this molecule, respectively). We infer that the ABCB5β signal detected at the cell periphery is not accounted for by the presence of this protein within the plasma membrane. It is possible that this signal results from the proximity between ER tubules and the plasma membrane, though not all of these ABCB5β-containing tubules were positive for calnexin.

Taken together, our findings support that ABCB5β is a microsomal protein, mostly found in the endoplasmic reticulum.

## 3. Discussion

The subcellular localization of ABCB5β, which appears to be the main ABCB5 isoform with putative transport activity expressed in melanoma cells [16], has remained unclear, primarily due to concerns about the reliability of anti-ABCB5 antibodies. Indeed, we (Figure 1) and others [19] found that commercial antibodies may be used to detect the overexpressed protein but fail to detect ABCB5 expressed endogenously in melanoma cells, at least by classical Western blotting and immunofluorescence methods. It may be that the protein expression level is simply below the detection level. Louphrasitthiphol et al. considered that the ABCB5 protein could exhibit a very long half-life. Hence, the temporal knockdown achieved through siRNA might not be sufficient to obtain a substantial reduction in ABCB5 protein expression. However, we observed a ≥80% decrease in *ABCB5* expression at the mRNA level up to 96 h post-transfection of the siRNAs, with no change in signal intensities in Western blotting experiments. Though it cannot be excluded, it seems unlikely that the ABCB5 half-life would be longer than 96 h. For instance, the half-life of ABCB1, which shares a ~55% identity with ABCB5FL and β, is approximately 26 h [21].

Several mammalian ABC transporters have been reported to localize at the plasma membrane and mediate the transportation of substances from inside the cell to the external environment [24]. However, approximately half of all ABC transporters are localized to intracellular compartments such as peroxisomes, lysosomes, and endosomes; endoplasmic reticulum; mitochondria; and Golgi apparatus [25]. As we could not rely on antibodies to analyze ABCB5β localization in the cells, we then made use of GFP- and HA-tagged proteins. All chimeras were found located in the endoplasmic reticulum, even when using a low expression promoter.

Newly synthesized membrane and secretory proteins must undergo correct folding and sometimes oligomerization for proper export from the endoplasmic reticulum. Polytopic membrane proteins, like ABC transporters, often encounter challenges in the folding process and/or trafficking from the ER. For instance, studies have shown that a substantial portion, up to 85%, of newly synthesized CFTR (wild-type ABCC7) undergoes degradation via the proteasome system [26]. Nonetheless, the correctly folded molecules effectively make their way to the plasma membrane to carry out their intended functions. In our study, we found that adding a N-ter GFP tag to ABCB9 (a half-transporter) did not prevent sorting, at least to some extent, to the lysosomes (which is the expected localization site for this protein according to [27,28]). Since ABCB5β was only detected in the ER of HeLa and MelJuSo cells, in all tested conditions, it is worth considering that this might be its main localization site under basal conditions.

Several ABC transporters are located, at least partly, in the ER membrane. They can exercise some important functions at this site. For instance, it has been reported that the ABCA9 transporter localizes to the ER in Hek293 cells and in some breast cancer cell lines and that it is involved in cholesterol import into this compartment [29]. When overexpressed, it decreases cholesterol synthesis due to its inhibitory effect on SREBP-2 activation and translocation to the nucleus, and it decreases breast cancer cell line proliferation. Within the ABCB subfamily, the ABCB2/B3 heterodimer associated with antigen processing resides in the ER. The heterodimer transports peptides from the cytosol into the endoplasmic reticulum (ER), thereby selecting peptides for binding to MHC class I molecules [30].

The function of ABCB5β remains elusive and the role of ABCB5β in melanoma has not been fully understood yet. Studies have investigated the potential role of ABCB5 in cancer multidrug resistance. However, most of them do not specify which isoform was investigated, and the data provided do not allow for discrimination between ABCB5FL and ABCB5β. This is a recurrent concern in many of the studies involving ABCB5 [1]. Keniya and colleagues have shown that ABCB5FL confers resistance to several drugs including anthracyclines as opposed to ABCB5β homodimers. Another study strictly focusing on ABCB5FL showed that this transporter mediates resistance to paclitaxel, docetaxel, and anthracyclines [14,17].

Conventional half transporters typically have only one NBD, either at the N- or C-terminal region. However, ABCB5β is predicted to possess a transmembrane domain (TMD) composed of six α-helices flanked by two intracellular NBDs (one of them complete, the other truncated). ABCB5β might form a dimer to create a functional transporter as potential dimerization motifs have been identified in its N-terminal region [31]. Whether homo- or heterodimers of ABCB5β (involving the association of this protein with other half-transporters of the ABCB family) confer drug resistance remains an open question.

Our experimental data indicate that ABCB5β expressed by itself localizes to the ER. An intriguing idea is that ABCB5β could travel to other compartments by becoming part of a heterodimer, similar to what was observed for the obligate ABCG5/G8 heterodimer. These two half transporters are dependent on one another for trafficking to the plasma membrane [32]. Under homodimer forms, they remain in the ER.

It has also been reported, for several transmembrane multimeric proteins, that heterodimerization can lead to the masking of ER retention signals. This is notably the case for the NMDA receptor, a molecule that is essential for neurotransmission. The major NR1 splice variant (NR1-1) and the NR2 subunits of NMDA are retained in the ER when expressed alone in heterologous cells and neurons, but when expressed together, they form functional receptors on the cell surface. These receptors are likely heterotetramers, assembled from NR1, NR2, and NR3 subunits [33].

Additionally, some proteins may attach themselves to carrier proteins to reach their final destination. It is notably the case of ABCD4, which localizes to the ER when expressed alone but traffics to LAMP1-positive compartments (late endosomes/lysosomes) when co-expressed with the lysosomal membrane protein LMBD1 [34].

In summary, we demonstrated that ABCB5β is predominantly localized to the ER even after treatment with SAHA, which restored the folding and increased the total level of ABCB5β expression to some extent. This discovery is the first step toward the elucidation of ABCB5β function in melanoma cells.

## 4. Materials and Methods

### 4.1. Antibodies

Three different polyclonal anti-ABCB5 antibodies were used in Western blotting analyses: rabbit anti-ABCB5 (Rockland Immunochemicals, Limerick, PA, USA, 600-401-A775, 1:500), rabbit anti-ABCB5 (Abcam, Cambridge, UK, ab80108, 1:100), and rabbit anti-ABCB5 (Atlas antibodies, Lund, Sweden, HPA026975, 1:500). We also used goat polyclonal anti-GFP (Rockland Immunochemicals, Limerick, PA, USA, 600-101-215, 1:1000); mouse monoclonal anti-α-Tubulin (Sigma-Aldrich, Hoeilaart, Belgium, T5168, 1:1000); rabbit polyclonal anti-GAPDH (Sigma-Aldrich, Hoeilaart, Belgium, G9545, 1:1000); mouse monoclonal anti-transferrin receptor (H68.4) (Invitrogen-Fisher Scientific, Brussels, Belgium, 1:500,13-6800). IRDye 680RD donkey anti-mouse secondary antibodies (Li-cor Biosciences, Lincoln, NE, USA, 926-68072, 1:10,000); IRDye 680RD goat anti-rabbit secondary antibodies (Li-cor Biosciences, Lincoln, NE, USA, 926-68071, 1:10,000); IRDye 800CW donkey anti-goat secondary antibodies (Li-cor Biosciences, Lincoln, NE, USA, 926-680724, 1:10,000). PageRuler™ Prestained Protein Ladder was purchased from Thermo Fisher Scientific, Hoeilaart, Belgium, 26617.

The following antibodies were used in immunofluorescence applications: rabbit polyclonal anti-ABCB5 (Rockland Immunochemicals, Limerick, PA, USA, 600-401-A775, 1:100); rabbit polyclonal anti-HA [C29F4] (Cell Signaling Technology, Leiden, The Netherlands, 3724, 1:100); rabbit anti-calnexin (Abcam, Cambridge, UK, ab22595, 1:100); rabbit anti-GM130 [EP892Y] (Abcam, Cambridge, UK, ab52649, 1:50); rabbit anti-TGN46 (ProteinTech, Manchester, UK, 13573-1-AP, 1:100); mouse anti-LAMP1 [H4A3] (DSHB, Iowa City, IA, USA, AB_2296838,1:50); mouse anti-melanoma [HMB45 + M2-7C10 + M2-9E3] (Abcam, Cambridge, UK, ab732, 1:50). We also used anti-mouse conjugated IgG (H + L) Cross-Adsorbed Secondary Antibody, Alexa Fluor™ 568 (Invitrogen-Fisher Scientific, Brussels, Belgium, A11004, 1:250), anti-rabbit Alexa Fluor 594 (Invitrogen-Fisher Scientific, Brussels, Belgium, A21207, 1:250), or Alexa Fluor 488 (Invitrogen-Fisher Scientific, Brussels, Belgium, A11034, 1:250).

### 4.2. Cell Culture, Transfection, and Treatment

The human melanoma cell line MelJuSo (ATCC, Molsheim, France, CVCL_1403) and HeLa cells (human cervix adenocarcinoma, ATCC Molsheim, France, CCL-2) were grown at 37 °C under 5% CO_2_ in DMEM with 4,5 g per L glucose, L-glutamine, sodium pyruvate (VWR, Leuven, Belgium, 392-0416) supplemented with 10% fetal bovine serum (FBS), 100 U/mL penicillin, and 100 µg/mL streptomycin (Lonza, Verviers, Belgium, DE17-602E). For transfection, cells were transfected with a mix containing Opti-Mem (Gibco, Looz, Belgium, 31985062), plasmid, and FuGENE^®^ HD Transfection Reagent (Promega, Leiden, The Netherlands, E2311) for HeLa cells and jetPRIME^®^ (Polyplus, Leuven, Belgium, 114-15) for MelJuSo cells, according to the manufacturer’s instructions. Experiments were performed 48 h after transfection unless otherwise specified. The inhibition of the proteasome was achieved by treating cells with 1 μM of MG132 (Sigma-Aldrich, Hoeilaart, Belgium, M7449) over 16 h. When indicated, the cells were transfected for 6 h and then treated overnight with 2.5 or 5 µM of SAHA (suberoylanilide hydroxamic acid).

### 4.3. RNA Interference

To silence ABCB5 expression, we purchased ONTARGETplus human ABCB5 siRNA SMARTpool from Dharmacon Inc. (Dharmacon, Horizon Discovery, Cambridge, UK, L-007303-01-0020) and transfected it with Lipofectamine™ RNAiMAX Transfection Reagent, according to the manufacturer’s instructions. ON-TARGET plus non-targeting pool (Dharmacon, Horizon Discovery, Cambridge, UK, D-001810-10-20) served as a reference point. Using a total final concentration of 90 nM, transfection of the siRNAs was performed at different times points of transfection from 24 h up to 96 h, where protein depletion efficiency was maximal, as shown by RT-qPCR.

### 4.4. RT-qPCR

Total RNA was isolated from MelJuSo cells using the NucleoSpin RNA plus kit (Machenerey-Nagel, Dueren, Germany). The quantity and quality of extracted RNA were assessed by spectrophotometry using the NanoDrop One/OneC Microvolume UV-Vis (Thermo Scientific, Dilbeek, Belgium). cDNA synthesis was performed using the RevertAid Minus First Strand cDNA synthesis kit (Thermo Scientific, Dilbeek, Belgium, K1631).

Specific primers for qPCR were designed to target *ABCB5FL* and *ABCB5β*, excluding the rest of the *ABCB5* isoforms (forward primer: 5’-GCAGATTTGATTGTGACCCT-3´; reverse primer: 5´-GACTCCATCTGTTCATCAGC-3´). Gene expression was determined using Takyon NoRox Sybr Mastermix Blue (Eurogentec, Liège, Belgium, UF-RTAD-D0701), according to the manufacturer’s recommendations on a BioRad CFX96 system (BioRad, Temse, Belgium). To normalize the expression analysis, Glyceraldehyde 3-phosphate dehydrogenase (GAPDH) (forward primer: 5´-ACCAGGTGGTCTCCTCTGAC-3´; reverse primer: 5´-TGCTGTAGCCAAATTCGTTG-3´) was used as a housekeeping gene.

### 4.5. Plasmid Constructs

The cDNA of the *ABCB5β* isoform (NP_ 848654.3) was inserted via BamHI and EcoRV in a pcDNA3.1(+)N-eGFP plasmid using an In-Fusion method (Takara Bio Inc., Saint-Germain-en-Laye, France, 638910). Similarly, the C-terminus tagged version of this construct was prepared by inserting *ABCB5β* in front of the GFP sequence using KpnI. HA-tagged constructs of ABCB5β were obtained by the insertion of a linker coding for HA via the NheI and KpnI restriction sites located upstream of the *ABCB5β* sequence in pcDNA3.1(+). Finally, *ABCB5β* was amplified by In-Fusion PCR with primers containing an EcoRI restriction site and inserted into an EcoRI digested pPGK plasmid (Addgene, Cambridge, MA, USA, plasmid # 35094) to obtain the HA-ABCB5β construct with a low expression promoter.

### 4.6. Western Blotting

Cell lysates prepared in a RIPA buffer (10 mM Tris-HCL, 1 mM EDTA, 1% Triton X-100, 0.1% sodium deoxycholate, 0.1% SDS, 140 Mm NaCl, 1× complete Mini protease inhibitor (Merck, Hoeilaart, Belgium, P5726)) were mixed with Laemmli’s sample buffer (with DTT) and resolved in an 8% SDS polyacrylamide gel. Proteins were transferred onto PVDF membranes (Immobilon^®^-FL, Sigma-Aldrich, Hoeilaart, Belgium) prior to the detection of the proteins of interest using the antibodies and dilutions listed above. Infrared signals were detected using an Odyssey infrared imaging system (LI-COR Biosciences, Lincoln, NE, USA).

Trichoplusia ni (high-five) insect cells were infected with a recombinant baculovirus carrying on the sequence of ABCB5β. Cells were harvested after 60–72 h of infection and stored at −80 °C. Total membrane vesicles were prepared with hypotonic lysis and differential centrifugation as detailed in [35]. Total membrane vesicles (1 µg) were heated for 30 min at 37 °C in 5× blue loading dye and resolved in an 8% SDS polyacrylamide gel, then transferred onto PVDF membranes (Immobilon-FL, Sigma-Aldrich, Hoeilaart, Belgium,) for 1 h 30 min at 110 V prior to the detection of the ABCB5 isoforms using the antibodies and dilutions listed above.

### 4.7. Biotinylation of Cell Surface Proteins

Transfected HeLa cells were washed twice with ice-cold PBS and five times with PBS supplemented with 0.7 mM CaCl2 and 0.25 mM MgSO4 (PBS++, pH 8). Cell-surface proteins were labeled by incubation with 1 mg/mL sulfo-NHS-SS-biotin [sulfosuccinimidyl 2-(biotinamido) ethyl-1,3-dithiopropionate] in PBS++ (Pierce, 79378) for 45 min on ice. Biotinylation was stopped by washing five times with ice-cold 50 mM glycine/PBS++ and cells were lysed with a RIPA buffer (50 mM Tris/HCl (pH 7.4), 120 mM NaCl, 1%(*v*/*v*) Triton X-100, 0.1%SDS, and 1%deoxycholate) containing protease inhibitors (Complete Mini Protease inhibitor cocktail tablets, Roche, Machelen, Belgium, 11836153001). Biotinylated proteins were precipitated with streptavidin–agarose beads (Pierce, Dilbeek, Belgium, 20353) by centrifugation at 4 °C for 1.5 min at 8000 rpm in a benchtop centrifuge and then eluted by incubation for 40 min at room temperature in Laemmli’s buffer containing 200 mM fresh DTT (dithiothreitol). One-tenth of the supernatant obtained after centrifugation, containing proteins unbound to streptavidin beads and proteins eluted from the beads (bound), were separated on SDS/PAGE (8% gel) and the proteins of interest were detected by Western blotting. Of note, GAPDH was detected as a control. As a cytosolic protein, it should be recovered in the non-biotinylated fraction.

### 4.8. Immunofluorescence

Cells were fixed in 4% paraformaldehyde for 10 min. Cells were then permeabilized with a solution of 0.2% Triton and 1% BSA (bovine serum albumin) in PBS for 10 min and blocked with 3% BSA. Then, the cells were incubated with primary and secondary antibodies diluted in BSA solution for 1 h. To visualize actin filaments, rhodamine-conjugated phalloidin (ThermoFisher, Dilbeek, Belgium, R415) was used. Lastly, coverslips were incubated for 10 min in DAPI (Merck, Hoeilaart, Belgium, 28718-90-3) to label nuclei and mounted with Mowiol mounting medium (Sigma-Aldrich, Hoeilaart, Belgium, 81381). Fixed samples were imaged with a Leica SP5 confocal microscope using a 40× or 63× objective (1.3 and 1.4 numerical aperture, respectively) or LSM 900 confocal microscope equipped with an Airyscan detector and with a Plan Apo 63× numerical aperture (NA) 1.4 oil immersion objective.

### 4.9. Image Analyses and Quantifications

All image quantifications were performed using (Fiji Is Just) ImageJ 1.54f (NIH, Bethesda, MY, USA). The quantification of colocalization between two channels was performed using the JACoP plugin in ImageJ software [36]. After setting a threshold for the signal of interest for each channel, the JACoP plugin was used to obtain the Mander’s coefficient and calculate the colocalization.

## Figures and Tables

**Figure 1 ijms-24-15847-f001:**
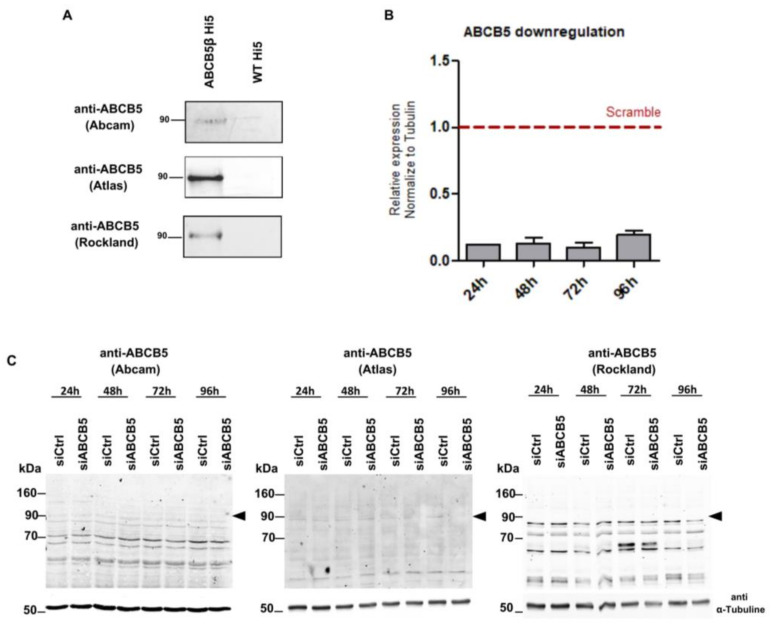
Analysis of several commercial anti-ABCB5 antibodies to analyze ABCB5 expression in MelJuSo cells. (**A**) Western blotting detection of human ABCB5β in membrane vesicles of high-five insect cells infected or not with *ABCB5β*-containing baculoviruses. Total membranes (1 µg) were resolved by SDS-PAGE prior to transfer on a PVDF membrane. Three different commercial anti-ABCB5 antibodies were tested: rabbit polyclonal anti-ABCB5 (Rockland Immunochemicals, Limerick, PA, USA, 600-401-A775), rabbit anti-ABCB5 (Abcam, Cambridge, UK, ab8010889), and rabbit anti-ABCB5 (Atlas antibodies, Lund, Sweden, HPA026975). (**B**) RT-qPCR analysis of *ABCB5* mRNA expression in MelJuSo cells after transfection with a scrambled siRNA pool (siCtrl) or a pool of 4 siRNAs directed against *ABCB5* (si*ABCB5*) over 24 h, 48 h, 72 h, or 96 h. Primers recognize a sequence located in the beta isoform. mRNA expression levels normalized to GAPDH and relative to siCtrl conditions are shown on the graph. (**C**) Western blotting detection of endogenous ABCB5 protein expression in MelJuSo cells transfected with siCtrl or si*ABCB5* over 24 h, 48 h, 72 h, and 96 h, using the 3 different commercial anti-ABCB5 antibodies anti-ABCB5 (Rockland Immunochemicals, Limerick, PA, USA, 600-401-A775), rabbit anti-ABCB5 (Abcam, Cambridge, UK, ab8010889), and rabbit anti-ABCB5 (Atlas antibodies, Lund, Sweden, HPA026975).

**Figure 2 ijms-24-15847-f002:**
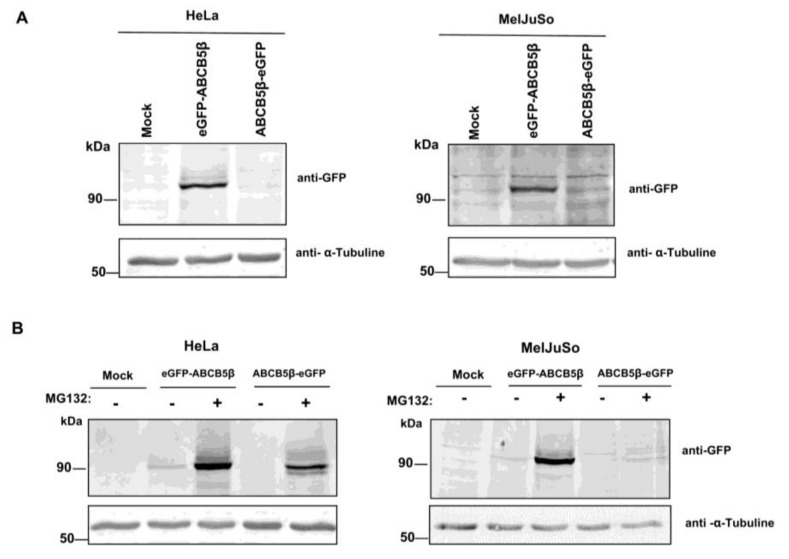
Expression of N-ter and C-ter GFP-tagged ABCB5β constructs in HeLa and MelJuSo cells. (**A**) pcDNA3.1 plasmids containing GFP-*ABCB5β* or *ABCB5β*-GFP cDNAs were transfected in HeLa or MelJuSo cells. A mock condition was included as a control (transfection with an empty pcDNA3.1 vector). Proteins were extracted 48 h post-transfection and resolved by SDS-PAGE. An anti-GFP antibody was used to detect the chimeric proteins. A-tubulin detection was used as a loading control. (**B**) HeLa and MelJuSo cells were transfected with GFP-*ABCB5β* and *ABCB5β*-GFP constructs for 48 h and treated for the last 6 h with 10 µM MG132. The proteins of interest were then detected as described in (**A**).

**Figure 3 ijms-24-15847-f003:**
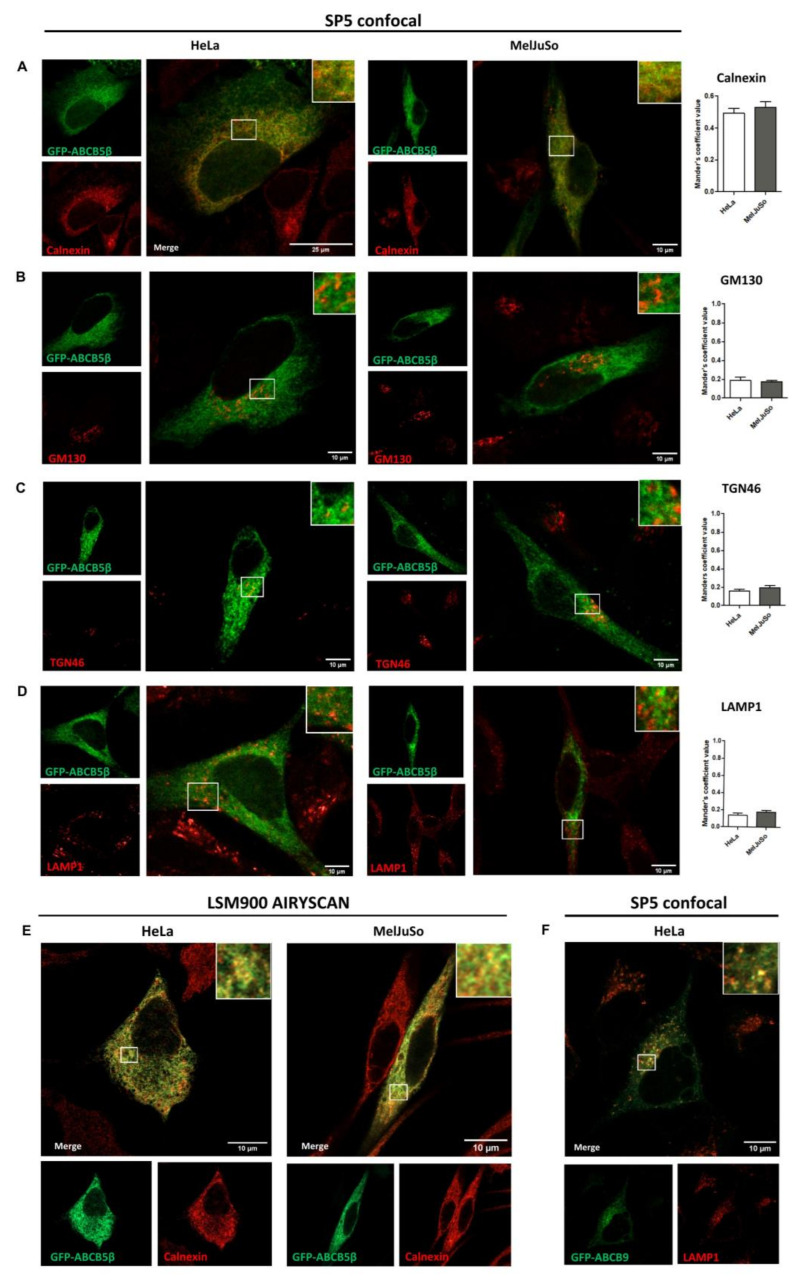
Subcellular localization of eGFP-ABCB5β in transfected HeLa and MelJuSo cells. After 48 h of transfection, cells were fixed with 4% of paraformaldehyde and processed for the detection of eGFP-ABCB5β (green) and different organelle markers (red): (**A**) calnexin for the ER, (**B**) GM130 for the cis-Golgi apparatus, (**C**) TGN46 for the trans-Golgi apparatus, (**D**) LAMP1 for late endosomes and lysosomes. Micrographs were obtained using an SP5 confocal microscope. Graphs show the quantification of co-localization extent by using Mander’s coefficient. *n* = 10 cells were analyzed from at least 3 independent experiments. (**E**) Higher resolution micrographs of ABCB5-calnexin co-localization taken using a Zeiss LSM900 Airyscan microscope. (**F**) Analysis of the presence of GFP-ABCB9 (with the tag in the N-ter position) in the lysosomes of HeLa cells detected using an anti-LAMP1 antibody. Imaging with SP5 Leica microscope. Scale bar = 25 or 10 µm. Magnified views in right insets.

**Figure 4 ijms-24-15847-f004:**
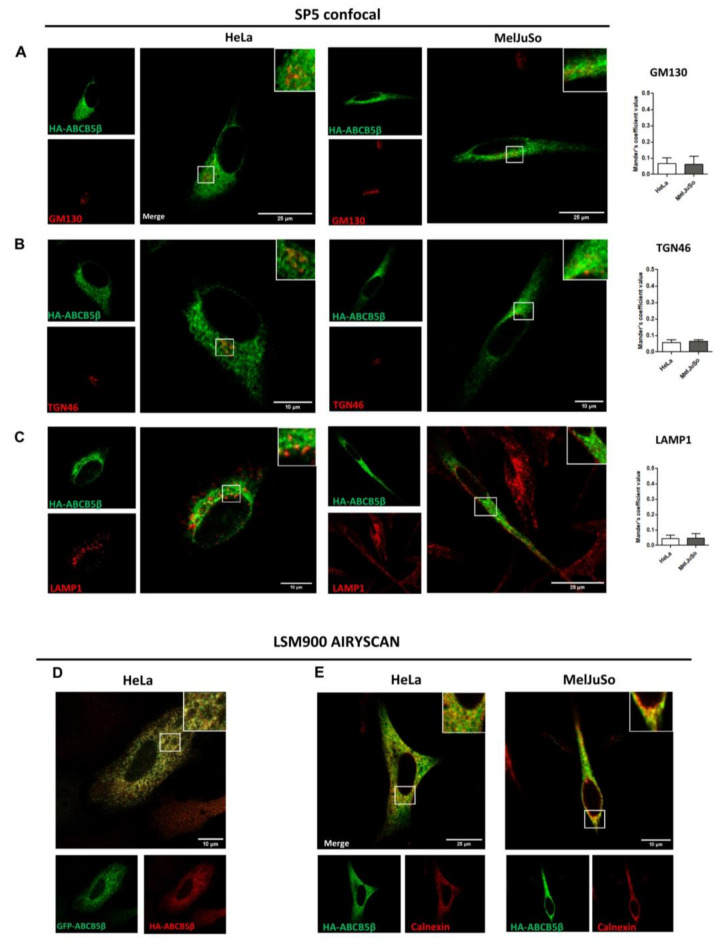
Subcellular localization of HA-ABCB5β in transfected HeLa and MelJuSo cells. (**A**–**C**) A total of 48 h post-transfection with HA-ABCB5β, HeLa, or MelJuSo cells were fixed with 4% of paraformaldehyde and processed for the detection of HA-ABCB5β (green) and different organelle markers (red): (**A**) GM130 for the cis-Golgi apparatus, (**B**) TGN46 for the trans-Golgi apparatus, (**C**) LAMP1 for late endosomes and lysosomes. Micrographs were obtained using an SP5 confocal microscope. Graphs show the quantification of co-localization extent by using Mander’s coefficient. *n* = 10 cells were analyzed from at least 3 independent experiments. (**D**) Analysis of co-localization between GFP-ABCB5β (green) and HA-ABCB5β (red, detected using an anti-HA antibody) with a Zeiss Airyscan microscope, 48 h after co-transfection in HeLa cells. (**E**) Higher resolution micrographs of HA-ABCB5—calnexin colocalization taken using a Zeiss LSM900 microscope. Scale bar = 25 or 10 µm. Magnified views in right insets.

**Figure 5 ijms-24-15847-f005:**
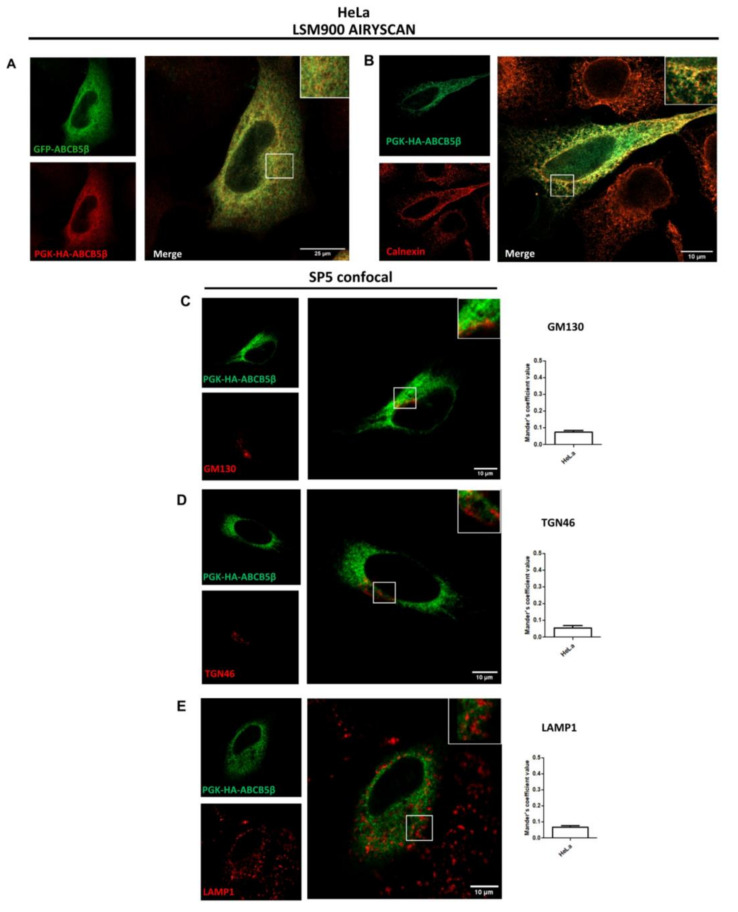
Subcellular localization of pPGK-HA-ABCB5β in transfected HeLa cells. (**A**) Colocalization analysis between GFP-ABCB5β (green) and PGK-HA-ABCB5β (red) detected with an anti-HA antibody (rabbit), 48 h post-transfection of HeLa cells. (**B**) Colocalization analysis between PGK-HA-ABCB5β (green) and calnexin. Micrographs in panels A and B were taken with a Zeiss LSM900 Airyscan microscope (high resolution). Additional colocalization analyses were conducted between PGK-HA-ABCB5β (green) and GM130 (**C**) and between TGN46 (**D**) and LAMP1 (**E**) (red). Graphs show the quantification of colocalization extent by using Mander’s coefficient. *n* = 10 cells were analyzed from at least 3 independent experiments. Scale bar = 25 or 10 µm. Micrographs in panels (**C**–**E**) were obtained using a SP5 confocal microscope. Magnified views in right insets.

**Figure 6 ijms-24-15847-f006:**
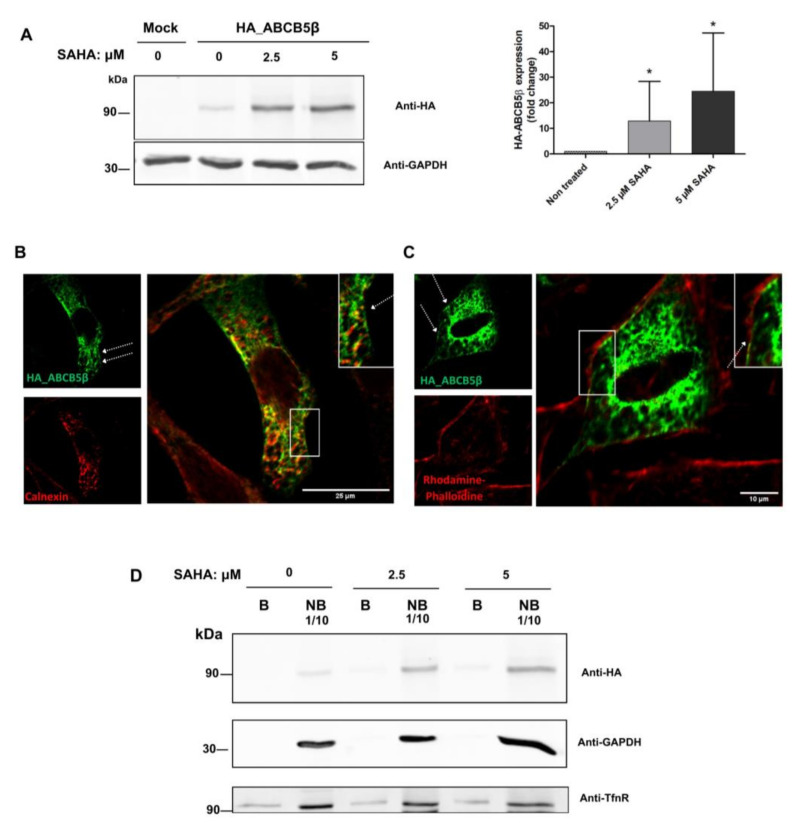
ABCB5β remained localized in the ER after treatment of the HeLa cells with SAHA. (**A**) Effect of an overnight SAHA treatment on the expression of HA-ABCB5β after transfection in HeLa cells. Proteins were extracted 24 h post-transfection and resolved by SDS-PAGE. An anti-HA antibody was used to detect the chimeric proteins. GAPDH detection was used as a loading control. The graph shows the quantification of expression in *n* = 4 independent experiments. * *p* < 0.05. (**B**) HeLa cells were transfected with HA-ABCB5β and treated overnight with SAHA (2.5 µM), then processed for the immunofluorescence detection of calnexin (**B**) or for detection of the actin cytoskeleton using Rhodamine-Phalloïdin (**C**). Scale bar = 25 or 10 µm. Micrographs obtained using a SP5 confocal microscope. A few cells exhibited ABCB5β labeling at the cell periphery (see arrows in panels B and C). Magnified views in right insets. (**D**) HeLa cells transfected with HA-ABCB5β were treated overnight with DMSO (control) or SAHA (2.5 or 5 µM). Proteins located at the cell surface were then biotinylated and separated from non-biotinylated (i.e., intracellular proteins) using streptavidin agarose beads. The presence of HA- ABCB5β was then analyzed in the B (biotinylated) and NB (non-biotinylated) fractions. Of note, 1/10th of the total NB fraction was loaded on the gel. GAPDH detection was used as a negative control. As a cytosolic protein, it is detected in the NB fraction. Detection of the transferrin receptor (TfnR) served as a positive control since this protein cycles between the plasma membrane and endosomes.

## Data Availability

All data used in this article are included within the figures and Appendix A files.

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
