# Peer review of "The β Isoform of Human ATP-Binding Cassette B5 Transporter, ABCB5β, Localizes to the Endoplasmic Reticulum"

_ijms, 2023, doi:10.3390/ijms242115847_

Round 1

Reviewer 1 Report

Comments and Suggestions for Authors

The manuscript titled "the β Isoform of Human ATP-Binding Cassette B5 Transporter, ABCB5β, is a Microsomal Protein" by Adriana Maria Díaz-Anaya et al described potential subcellular localization of  ABCB5β, a protein frequently mutated in melanoma with unclear function. The authors used exprssed tagged protein and markers of subcelluar compartments and concluded that  ABCB5β was located in microsomes -- which is in the title.  Treatment with SAHA, a molecule that promotes 17 chaperone-assisted folding, increased the expression of  tagged ABCB5β expression. 

The strength: The results are interesting to the research communities. The authors evaluated three ABCB5β antibodies and showed that they were not able to detect the endogenous level of ABCB5β in cells, hence they used tagged proteins (GFP or HA, with HA, strong and weak promoters are used). 

Weakness:

the title and the conclusion: a microsomal protein, does it mean it is located in ER? if so, why not say ER? microsomes are not organelles.

The  signals for marker LAMP1 are quite weak -- lapm1 is a highly expressed protein and the signal can be stronger. It does seem that in figures LAMP1 showed some overlapping with ABCB5β signal -- is that true?

Figure 1D is unnecessary as Fig1C already showed that the Rockland antibody detected a whole bunch of non-specific proteins. 

Reviewer 2 Report

Comments and Suggestions for Authors

The manuscript successfully determined the subcellular localization of ABCB5β, a member of the ABC transporter superfamily, which was cloned from melanocytes using a series of in vitro experiments. The study's rationale, focused on uncovering the underlying mechanisms and the extent to which ABCB5 isoforms contribute to melanoma tumorigenesis, is clearly articulated.

The methodology employed in the experiment, including cell culture with transfection and treatment, Western blotting, immunohistochemical analysis, RT-qPCR, and RNA interference, is notably sophisticated. The results, demonstrating that ABCB5β, when expressed on its own, localizes to the endoplasmic reticulum (ER), provide new insights supported by robust evidence. This finding represents an important initial step in understanding its function in melanoma cells.

Author Response

We thank you for your time and for the very positive feedback. 
